# The Silent Threat: A Retrospective Cohort Study on the Impact of Prediabetes on ST-Elevation Myocardial Infarction Survival—A Call to Action!

**DOI:** 10.3390/biomedicines12102223

**Published:** 2024-09-29

**Authors:** Elke Boxhammer, Matthias Hammerer, Christiana Schernthaner, Erika Prinz, Mathias C. Brandt, Michael Lichtenauer, Alexander E. Berezin, Wilfried Wintersteller, Uta C. Hoppe, Kristen L. Kopp

**Affiliations:** Department of Internal Medicine II, Division of Cardiology, Paracelsus Medical University of Salzburg, 5020 Salzburg, Austria

**Keywords:** cardiovascular disease, diabetes, glycemic status, prediabetes, STEMI

## Abstract

**Background:** Prediabetes is frequently underdiagnosed and undertreated, yet it poses significant cardiovascular risks. This study investigates the impact of prediabetes on short- and long-term survival outcomes in patients who experienced ST-elevation myocardial infarction (STEMI). **Methods:** In this retrospective, single-center cohort study, we evaluated 725 STEMI patients stratified into non-diabetic, prediabetic, and diabetic groups based on HbA1c levels at presentation. A Kaplan–Meier survival analysis was employed to compare long-term outcomes over a three-year follow-up period. Cardiovascular risk factors, including hypertension and dyslipidemia, were analyzed across the groups. The discriminatory power of HbA1c for predicting all-cause mortality was assessed using an Area Under the Receiver Operating Characteristic (AUROC) analysis. **Results**: Of the 725 patients, 407 (56.1%) were non-diabetic, 184 (25.4%) were prediabetic, and 134 (18.5%) were diabetic. Prediabetic patients exhibited significant additional cardiovascular risk factors, such as arterial hypertension (67.4%) and dyslipidemia (78.3%), with prevalence rates between those of non-diabetic and diabetic patients. The Kaplan–Meier analysis revealed that at a three-year follow-up, prediabetic patients faced a survival disadvantage, with a significant decrease in survival rates compared to non-diabetic patients (log-rank *p* = 0.016); their survival outcomes approached those of diabetic patients (*p* = 0.125). The AUROC analysis demonstrated that HbA1c was a significant predictor of short- and long-term mortality, with a cut-off value of 5.75% and an Area Under the Curve (AUC) of 0.580–0.617 (95% CI: 0.458–0.721), indicating a moderate ability to predict survival in patients with STEMI. **Conclusions:** Prediabetes significantly worsens survival outcomes following STEMI, nearly approaching the risk level of diabetes. Integrating rigorous cardiovascular risk management strategies for prediabetic individuals, including lifestyle interventions and potentially pharmacological treatments, could prevent the progression to diabetes and mitigate associated cardiovascular risks.

## 1. Introduction

Acute ST-segment elevation myocardial infarction (STEMI) stands as a formidable medical emergency, demanding swift intervention and comprehensive management. Despite advancements in acute care, long-term prognosis remains a challenge, particularly in patients with underlying metabolic disorders [1,2]. Among these, prediabetes emerges as a critical, yet often underappreciated, factor influencing survival outcomes [3,4].

Prediabetes, which is defined by impaired fasting glucose or impaired glucose tolerance, represents an intermediate metabolic state with blood glucose levels elevated above normal but not high enough to be classified as diabetes. This condition is alarmingly prevalent, with significant implications for cardiovascular health [5]. The pathophysiological mechanisms linking prediabetes to adverse cardiac events are multifaceted, involving insulin resistance, subclinical inflammation, endothelial dysfunction, and accelerated atherogenesis [6].

The current literature highlights a compelling connection between prediabetes and increased cardiovascular risk, comparable to that observed in overt diabetes [7]. Patients with prediabetes are at a higher risk of progressing to diabetes and experiencing cardiovascular events, including STEMI. Importantly, evidence suggests that prediabetic individuals who survive a STEMI face a significantly elevated long-term risk of mortality and recurrent cardiac events compared to normoglycemic patients [8].

The prognostic importance of prediabetes in the context of STEMI is underscored by several key studies. Research has demonstrated that even modest elevations in glycemic indices, such as hemoglobin A1c (HbA1c), are associated with worse outcomes post-STEMI. This highlights the insidious impact of prediabetes on myocardial recovery and long-term survival, necessitating the early identification and aggressive management of this at-risk population [9].

In clinical practice, the challenge lies in recognizing prediabetes as a potent risk factor and integrating it into risk stratification models for STEMI patients. Traditional risk assessments often overlook the subtle yet significant influence of prediabetic states, potentially leading to suboptimal management strategies [10]. 

Incorporating glycemic parameters such as HbA1c into predictive models may enhance the precision of risk stratification, enabling tailored therapeutic approaches that address the unique metabolic vulnerabilities of prediabetic patients. Therefore, our study aims to bridge this gap by providing robust evidence on the impact of prediabetes on survival following STEMI. By analyzing a large cohort and employing rigorous statistical methods, we seek to elucidate the prognostic significance of prediabetes. This endeavor not only advances our understanding of the interplay between glycemic status and cardiovascular outcomes but also paves the way for personalized intervention strategies designed to improve long-term survival in STEMI patients.

## 2. Materials and Methods

### 2.1. Study Population

For this retrospective study, all patients (*n* = 964) presenting with STEMI at a single large tertiary center in Salzburg, Austria, between 1 January 2018, and 31 December 2020, were screened. Eligible participants were STEMI patients aged 18 years or older, with documented cardiovascular risk factors and available baseline HbA1c values obtained during their initial hospitalization for STEMI (*n* = 725). Patients lacking HbA1c or with conservative treatment (without receiving a percutaneous coronary intervention) were excluded from this study.

The primary endpoint for this study was overall mortality within a three-year follow-up period. To determine outcomes, we established clear criteria for classifying mortality, which included reviewing medical records and death certificates. Additionally, we monitored and recorded cardiovascular events, defining them based on clinical guidelines and patient follow-up data. This thorough approach ensured an accurate assessment of survival rates and provided a comprehensive understanding of the long-term outcomes in this patient cohort. 

To estimate the required sample size for this study, we utilized G*Power 3.1: (University of Duesseldorf, Duesseldorf, Germany) for an a priori analysis. The calculations indicated that each group should consist of 105 participants to ensure adequate statistical power. This estimation was based on an expected effect size of 0.5, an alpha level set at 0.05, and a target power of 0.95, with an equal distribution of participants across the study groups.

### 2.2. Ethics Declaration

This study was approved by the State of Salzburg Ethics Commission (EK-Nr. 1038/2021). All data handling complied with the Declaration of Helsinki principles and Good Clinical Practice (ICH-GCP) guidelines.

### 2.3. Data Extraction

Information was collected from the ORBIS electronic medical records system (Agfa-ORBIS; Agfa HealthCare GmbH, Mortsel, Belgium, Version 08043301.04110DACHL) and the medical archiving system (Krankengeschichtsarchiv, Uniklinikum Salzburg, Softworx by Andreas Schwab TM, 2008, Salzburg, Austria) at the University Clinic Salzburg in Austria. This data included patient charts, as well as reports from admissions, discharges, and laboratory results throughout the course of treatment. Subsequently, patient data were pseudo-anonymized and compiled into an Excel ((2016). Microsoft Excel, Redmond, WA, USA) database for analysis. 

### 2.4. Laboratory Chemical Examinations

All laboratory values were analyzed at the University Institute for Medical-Chemical Laboratory Diagnostics at University Hospital Salzburg. HbA1c levels were measured using high-performance liquid chromatography (HPLC), which is a standard and precise method for quantifying the percentage distribution of glycosylated hemoglobin. The intra- and inter-assay coefficients of variation for all kits were found to be less than 10%. This method allows for accurate assessment of average blood glucose levels over the preceding 2–3 months. The Friedewald formula was used to calculate plasma low-density lipoprotein cholesterol (LDL-C) concentration when triglyceride levels were below 275 mg/dL. For higher triglyceride levels, the LDL particle number was assessed using the c702 module of the Roche Cobas^®^ 8000 analyzer (Roche Diagnostics, Mannheim, Germany), adhering to the manufacturer’s guidelines for measuring LDL-C levels.

### 2.5. Clinical Criteria of STEMI

STEMI was diagnosed based on guidelines set by the European Society of Cardiology (ESC) [11]. 

### 2.6. Categorization of Glycemic Status

Patients were categorized into three groups based on their glycemic status using the following two-step approach:

Step 1: Medical History Review

Initially, a comprehensive medical history review (anamnesis) was conducted for each patient to determine if they had a prior diagnosis of diabetes and how it was managed (diet only, oral medication, or insulin). If a patient was identified as having pre-existing diabetes, they were categorized as diabetics regardless of their current HbA1c levels.

Step 2: HbA1c Measurement

For patients without a prior diabetes diagnosis, HbA1c levels were measured and used to classify them according to the following criteria:

Non-diabetic: HbA1c < 5.7%;

Prediabetic: HbA1c 5.7–6.4%;

Diabetic (new diagnosis): HbA1c ≥ 6.5%.

Therefore, the diagnosis of non-diabetic and prediabetic patients was based solely on the measured HbA1c at the time of the STEMI event, whereas patients with a prior diagnosis of diabetes were categorized based on their medical history, independent of their current HbA1c levels [12]. 

### 2.7. Comorbidity Identification

Obesity, chronic kidney disease (CKD), heart failure, hypertension, and dyslipidemia were defined and evaluated according to current clinical guidelines [13,14,15,16]. The conventional CKD-EPI formula was used to estimate the glomerular filtration rate (eGFR).

### 2.8. Statistical Analyses

Statistical analyses and data visualization were performed using SPSS Version 25 (IBM SPSS Statistics, Armonk, New York, NY, USA). The Kolmogorov–Smirnov–Lilliefors test was employed to evaluate the normality of variable distributions. Metrics exhibiting normal distribution were reported as mean ± standard deviation (SD) and analyzed using an unpaired Student’s *t*-test. For metrics not following a normal distribution, data were presented as the median and interquartile range (IQR), with statistical comparisons made using the Mann–Whitney U test. Categorical variables were expressed as frequencies and percentages, with comparisons conducted using the chi-square test. Kaplan–Meier survival curves and corresponding log-rank tests were performed for both short- and long-term survival. Further analyses included calculating the area under the receiver operating characteristic (AUROC) curves for the studied HbA1c values in relation to short-term (up to 6 months) and long-term survival (up to 3 years). This also involved determining cut-off values with corresponding sensitivity, specificity, and the Youden index (YI). In this study, we utilized various *p*-values to compare the survival rates among different glycemic groups. The *p*_general value represents the overall comparison among all three groups (non-diabetic, prediabetic, and diabetic). The *p** value denotes the comparison between non-diabetic and prediabetic patients. The *p*** value signifies the comparison between non-diabetic and diabetic patients, while the *p**** value represents the comparison between prediabetic and diabetic patients. A *p*-value of ≤0.05 was determined as statistical significance.

## 3. Results

### 3.1. Study Population—No Diabetes, Prediabetes, and Diabetes

The study population comprised 725 patients presenting with STEMI at a single, large tertiary care center in Salzburg, Austria. These patients were systematically categorized based on their diabetes status. Initially, we screened for a history of diabetes. Of the total, 107 patients (14.8%) had a prior diabetes diagnosis. We reviewed their medical history to determine the nature of their diabetes management, identifying whether they controlled their condition through diet (*n* = 14, 13.1%), oral medication (*n* = 62, 57.9%), or insulin ± oral medication (*n* = 31, 29.0%). Patients falling into these categories were classified as having pre-diagnosed diabetes, regardless of their HbA1c levels.

For the remaining 618 patients (85.2%) without a known history of diabetes, we assessed their HbA1c levels to determine their glycemic status at the time of the STEMI event. Based on the HbA1c values, patients were categorized into three groups. Those with HbA1c levels below 5.7% (*n* = 407, 65.9%) were classified as non-diabetic, those with HbA1c levels between 5.7% and 6.4% (*n* = 184, 29.8%) were considered prediabetic, and those with HbA1c levels equal to or greater than 6.5% (*n* = 27, 4.4%) were identified as having newly diagnosed diabetes. This detailed categorization is visualized in Figure 1.

### 3.2. Baseline Characteristics

In this study, we analyzed the baseline characteristics (Table 1) of the following three groups: no diabetes, prediabetes, and diabetes. A demographic analysis revealed no significant gender differences among the groups. However, age distribution indicated that non-diabetic patients were generally younger, with a higher percentage under 50 years compared to their prediabetic and diabetic counterparts, with significant differences between non-diabetic and diabetic patients (*p*** < 0.001) and non-diabetic and prediabetic patients (*p** = 0.045). The comparison between prediabetic and diabetic patients also showed significant differences (*p**** = 0.013).

Cardiovascular risk factors showed notable variations. Arterial hypertension was more prevalent in the diabetic group (82.8%) compared to non-diabetic (64.1%, *p*** < 0.001) and prediabetic (67.4%, *p** = 0.034) patients, with a significant difference also observed between prediabetic and diabetic patients (*p**** = 0.021). Dyslipidemia was most common in the prediabetic group (78.3%), and smoking rates were relatively similar across all groups, though slightly lower in the diabetic group (55.2%).

A body mass index (BMI) analysis highlighted significant differences, with diabetic patients having higher BMI values. The prevalence of obesity (BMI ≥ 30 kg/m^2^) was particularly pronounced in the diabetic group, with 27.6% having a BMI in the 30.0–34.9 kg/m^2^ range and 9.7% in the 35.0–39.9 kg/m^2^ range.

Regarding STEMI characteristics, diabetic patients exhibited a higher incidence of three-vessel disease (44.0%) compared to non-diabetic (30.7%, *p*** < 0.001) and prediabetic (25.0%, *p** = 0.015) patients. Cardiogenic shock was also more common in the diabetic group (20.9%). Laboratory values further illustrated significant variations, particularly in glycemic control markers. Lipid profiles indicated lower high-density lipoprotein cholesterol (HDL) and higher triglyceride levels in diabetic patients.

### 3.3. Kaplan–Meier Curves—No Diabetes, Prediabetes, and Diabetes

Figure 2 presents Kaplan–Meier survival curves for the study population stratified by glycemic status (non-diabetic, prediabetic, and diabetic) over a three-year period following STEMI. Survival rates were analyzed at 30 days, 90 days, 180 days, 1 year, 2 years, and 3 years.

At 30 days, survival rates did not significantly differ between the groups (log-rank *p* = 0.184). By 90 days, survival rates showed a trend toward significance (log-rank *p* = 0.065), with a significant difference between non-diabetic and diabetic patients (*p*** = 0.019). At 180 days, differences became more pronounced (log-rank *p* = 0.001), with significant disparities between non-diabetic and diabetic (*p*** < 0.001) and prediabetic and diabetic patients (*p**** = 0.037).

A one-year survival analysis showed significant differences (log-rank *p* = 0.002), mainly between non-diabetic and diabetic patients (*p*** < 0.001), with prediabetic vs. diabetic comparisons nearing significance (*p**** = 0.061). At two years, significant differences persisted (log-rank *p* = 0.001), with notable differences between non-diabetic and diabetic (*p*** < 0.001), and prediabetic and diabetic patients (*p**** = 0.083).

At three years, survival rates diverged significantly (log-rank *p* < 0.001), highlighting the impact of diabetes on long-term survival. The non-diabetic vs. diabetic (*p*** < 0.001) and prediabetic vs. diabetic (*p**** = 0.125) comparisons underscored the risk. The non-diabetic vs. prediabetic comparison also became significant (*p** = 0.016), indicating substantial long-term risks for prediabetic patients.

### 3.4. AUROC Analysis of HbA1c for Predicting Mortality

Figure 3 illustrates the AUROC analysis for HbA1c as a predictor of mortality at various time intervals, as follows: 30 days, 90 days, 180 days, 1 year, 2 years, and 3 years post-STEMI. The analysis shows that HbA1c has a moderate predictive power for mortality, with AUROC values improving slightly over longer follow-up periods. A HbA1c cut-off value of 5.75%, and therefore prediabetic glycemic status, was consistently identified as relevant across all time frames, demonstrating sensitivities between 57 and 61% and specificities between 66 and 68%. This highlights the utility of HbA1c in identifying patients at increased risk of mortality following STEMI.

### 3.5. Study Population—No Diabetes, Prediabetes, and Diabetes with Further Cardiovascular Risk Factors

Figure 4 presents a comprehensive flowchart that illustrates the distribution of 725 patients diagnosed with STEMI, categorized by their glycemic status and associated comorbid conditions.

Among the total number of STEMI patients classified as having no diabetes, 64.1% (*n* = 261) suffered from hypertension, with 59.4% (*n* = 155) of these hypertensive patients receiving pretreatment. Additionally, 62.9% (*n* = 256) were smokers. Furthermore, 70.5% (*n* = 287) of non-diabetic patients had dyslipidemia, and 15.7% (*n* = 45) of these dyslipidemic patients were pretreated.

Within the prediabetic subgroup, 67.4% (*n* = 124) exhibited hypertension, and 62.9% (*n* = 78) of those hypertensive patients had been pretreated. With respect to smoking, 61.4% (*n* = 113) of prediabetic patients were smokers. Regarding dyslipidemia, 78.3% (*n* = 144) of prediabetic patients had this condition and 24.3% (*n* = 35) were pretreated.

The diabetic group included 82.8% (*n* = 111) of patients with hypertension, with 76.6% (*n* = 85) of these hypertensive patients being pretreated. Additionally, 54.8% (*n* = 74) were smokers. Regarding dyslipidemia, 69.4% (*n* = 93) of diabetic patients were affected, with 45.2% (*n* = 42) being pretreated.

### 3.6. Kaplan–Meier Curves—No Diabetes, Prediabetes, and Diabetes with Further Cardiovascular Risk Factors

The following Kaplan–Meier survival curves presented in Figure 5 illustrate the overall survival of STEMI patients, stratified by glycemic status (no diabetes, prediabetes, and diabetes) in conjunction with the presence of arterial hypertension (Figure 5A), smoking (Figure 5B), and dyslipidemia (Figure 5C). Key survival probabilities with respective *p*-values at specific time points (30 days, 90 days, 180 days, 1 year, 2 years, and 3 years) highlight the significance of differences in survival among the groups. Each figure also includes a tabular overview of the numbers at risk at these specific time points, providing a comprehensive view of the patient population throughout the study period.

#### 3.6.1. Glycemic Status + Arterial Hypertension

The Kaplan–Meier survival curve in Figure 5A shows significant differences in survival rates among non-diabetic, prediabetic, and diabetic STEMI patients over three years, with arterial hypertension as a stratifying factor. At 30 days, survival differences were not significant overall (*p*_general = 0.148), although a trend toward lower survival in diabetic patients was observed (*p** = 0.063). By 90 days, differences approached significance (*p*_general = 0.077), with significantly lower survival in diabetic patients compared to non-diabetic patients (*p*** = 0.034). At 180 days, significant differences were evident (*p*_general = 0.001), particularly between non-diabetic and diabetic patients (*p*** < 0.001). A one-year survival analysis revealed significant disparities (*p*_general = 0.002), with diabetic patients showing worse outcomes (*p*** < 0.001). By two years, the differences remained significant (*p*_general = 0.001), with poorer survival in diabetic patients compared to non-diabetic (*p*** < 0.001) and prediabetic patients (*p** = 0.006). At three years, the survival disparities were highly significant (*p*_general < 0.001), with diabetic patients continuing to show the lowest survival rates (*p*** < 0.001) and significant differences observed between non-diabetic and prediabetic patients (*p** = 0.005).

#### 3.6.2. Glycemic Status + Smoking

The Kaplan–Meier survival curves for STEMI patients show significant differences based on glycemic status and smoking over three years (Figure 5B). At 30 days, no significant survival differences were observed (*p*_general = 0.419). By 90 days, the survival differences remained non-significant overall (*p*_general = 0.581). At 180 days, the differences approached significance (*p*_general = 0.149), with a notable trend between non-diabetic and diabetic patients (*p*** = 0.061). A one-year analysis showed significant differences, with diabetic patients having worse survival compared to non-diabetic patients (*p*** = 0.036). Two-year survival differences remained significant between non-diabetic and diabetic patients (*p*** = 0.035), with overall differences nearing significance (*p*_general = 0.111). By three years, significant survival disparities were observed (*p*_general = 0.040), especially between non-diabetic and diabetic patients (*p*** = 0.011).

#### 3.6.3. Glycemic Status + Dyslipidemia

The Kaplan–Meier survival curves for STEMI patients, stratified by glycemic status and dyslipidemia, reveal significant differences in survival rates over three years (Figure 5C). At 30 days, survival differences were not significant (*p*_general = 0.236), with non-significant comparisons between the different groups. By 90 days, overall survival differences remained non-significant (*p*_general = 0.121), although the difference between non-diabetic and prediabetic patients approached significance (*p** = 0.054). At 180 days, the survival differences approached significance overall (*p*_general = 0.123), again with a notable trend between non-diabetic and diabetic patients (*p*** = 0.065). A one-year survival analysis indicated a trend toward significant differences (*p*_general = 0.112), particularly between non-diabetic and prediabetic patients (*p** = 0.057). By two years, significant survival differences emerged overall (*p*_general = 0.027), with significant differences between non-diabetic and prediabetic patients (*p** = 0.012) and non-diabetic and diabetic patients (*p*** = 0.043). At three years, survival disparities were highly significant (*p*_general < 0.001), especially between non-diabetic and prediabetic patients (*p** < 0.001) and non-diabetic and diabetic patients (*p*** = 0.001).

## 4. Discussion

### 4.1. The Overlooked Risk in Prediabetic Patients

This study highlights the following critical and often overlooked aspect of cardiovascular care on long-term survival following acute STEMI: the substantial risk faced by prediabetic patients, a group that is frequently underdiagnosed and undertreated. Our findings reveal that prediabetic individuals presenting with STEMI exhibit survival outcomes that are markedly worse than those of their non-diabetic counterparts and, in some cases, approach the poor outcomes observed in diabetic patients. This raises the following urgent question for the medical community: are we doing enough for prediabetic patients?

### 4.2. Baseline Characteristics and Cardiovascular Risk

The baseline characteristics of our cohort underscore the increasing prevalence of dysglycemia among patients with cardiovascular disease. Prediabetic patients, while not yet reaching the glycemic thresholds for a diabetes diagnosis, exhibited significant cardiovascular risk factors, such as arterial hypertension and dyslipidemia [17], and are medically underserved [18]. These risk factors were present at levels intermediate between those observed in non-diabetic and diabetic patients, suggesting a continuum of risk that starts well before the onset of overt diabetes [19,20].

### 4.3. Kaplan–Meier Survival Analysis: A Wake-Up Call

The Kaplan–Meier survival curves vividly illustrate the neglected plight of prediabetic patients. While the immediate post-STEMI survival (30 days) was comparable across all groups, as also described in Tian et al. [21], the divergence in survival rates became pronounced with longer follow-ups. By 90 days and 180 days, prediabetic patients began to show significantly worse survival outcomes, and this trend continued to worsen over one, two, and three years. By the three-year mark, prediabetic patients faced a survival disadvantage nearly as severe as that faced by diabetic patients, particularly when considering the compounded effects of other cardiovascular risk factors, such as arterial hypertension and dyslipidemia. These findings are consistent with those of Xu et al. [17,22], who also reported that prediabetes was associated with worse long-term outcomes in young patients ≤ 45 years of age with acute coronary syndrome. Another point worth discussing is the proportion of STEMI patients with complete revascularization. Unfortunately, it is not possible to assess the contribution of this factor to the survival rate of prediabetic patients compared to diabetic patients.

### 4.4. Better Management of Diabetes and Cardiovascular Risk Factors (Before or after STEMI)

This study also highlights the importance of better management of diabetes, particularly concerning additional cardiovascular risk factors, such as hypertension and dyslipidemia, as risk-based atherosclerotic cardiovascular disease (ASCVD) prevention deficits were observed in patients with diabetes and prediabetes at the time of presentation for STEMI. The effective treatment of diabetes, including optimal control of blood pressure and lipid levels, has been shown to significantly reduce cardiovascular events and improve survival outcomes [17,23]. This comprehensive approach to managing diabetes could serve as a model for treating prediabetic patients, addressing not only glycemic control but also the broader spectrum of cardiovascular risk factors. By adopting a holistic approach that addresses these additional risk factors, healthcare providers can significantly enhance patient outcomes and reduce the burden of cardiovascular disease in both prediabetic and diabetic populations.

### 4.5. The Imperative for Change: Treating Prediabetes

Our findings compel a re-evaluation of current clinical practices and treatment guidelines. The data clearly indicate that prediabetic patients are not merely in a transitional state but are at a critical juncture, at which early intervention could significantly alter their cardiovascular outcomes. Currently, the standard of care predominantly focuses on the treatment and management of diabetes, with prediabetic patients often receiving minimal intervention. Indeed, the ESC Guidelines for the Management of Hyperlipidemia, for example, recognized diabetes mellitus I and II patients as those at high or very high risk for fatal 10-year cardiovascular disease (CVD), depending on the constellation of further risk factors, including hypertension, hyperlipidemia, smoking status, and duration of diabetes; however, prediabetes is not yet found in higher risk categories [16,24]. This oversight places prediabetic patients at a disproportionate risk, as evidenced by their survival rates in this study. Similar concerns were highlighted by Kim et al. [25], who found that prediabetic patients with STEMI and multivessel disease had worse clinical outcomes compared to non-diabetic patients, underscoring the need for more aggressive management in this population.

### 4.6. A Call to Action: Revolutionizing Prediabetes Management

The evidence presented necessitates a paradigm shift in how we approach prediabetes. It is imperative that we extend rigorous cardiovascular risk management strategies to prediabetic patients, similar to those employed for diabetic patients [26]. This includes aggressive monitoring (HbA1c, fasting glucose, oral glucose tolerance test), lifestyle interventions, and, potentially, pharmacological treatments aimed at controlling glycemic levels, blood pressure, and lipid profiles. The integration of prediabetic management into routine cardiovascular care could be transformative, while there are controversial findings regarding the inconclusive impact of the agents, which are widely used in prediabetes/diabetes mellitus type II, i.e., Sodium-glucose cotransporter 2 (SGLT-2) inhibitors, on the clinical course of STEMI [27]. The proactive treatment of prediabetes has the potential to prevent the progression to diabetes and mitigate the associated cardiovascular risks [28]. The establishment of comprehensive prediabetes management programs could reduce the incidence of STEMI and improve long-term survival outcomes for this vulnerable population.

## 5. Limitation

This study has several limitations that must be considered when interpreting the results.

Firstly, the single-center retrospective design may limit the generalizability of our findings to other populations and healthcare settings.

Secondly, the classification of glycemic status was based on a single HbA1c measurement, which may not capture fluctuations in glycemic control over time. Current guidelines recommend conducting more than one test to confirm the diagnosis, and using only a single measurement may lead to misclassification, thereby impacting this study’s validity. Misclassification can obscure the true prevalence of prediabetes and its associated risks, potentially skewing the results and conclusions drawn.

Thirdly, potential confounding factors, such as variations in treatment regimens and patient adherence to medication and lifestyle modifications, were not addressed during the follow-up period. Yet, we did not evaluate the impact of several confounding factors, such as atrial fibrillation and the severity of atherosclerosis, left ventricular hypertrophy (LVH), and heart failure and its phenotypes, which had been previously defined as powerful predictors for all-cause and cardiovascular (CV) mortality in STEMI, on the risk of the STEMI-related outcomes.

Fourthly, the follow-up period, although extending to three years, may still be insufficient to fully capture the long-term impact of prediabetes on survival outcomes post-STEMI. Future multi-center studies with longer follow-up periods and more comprehensive assessments of glycemic control and cardiovascular risk factors are warranted to validate and extend our findings for post-STEMI patients, as well as other patient populations, in secondary and primary cardiovascular prevention.

## 6. Conclusions

In conclusion, this study highlights the critical need to reconsider our approach to cardiovascular risk management by prioritizing prediabetic patients. The significant survival disparities identified through a Kaplan–Meier analysis underscore the importance of early intervention in prediabetic individuals with STEMI. While the findings suggest that treating prediabetes with comparable seriousness to diabetes may be associated with improved patient outcomes, further validation through more robust data analysis is necessary to substantiate these claims.

To optimize care, we propose exploring multi-marker strategies (HbA1c, fasting glucose, oral glucose tolerance test) that could enhance the predictive value of interventions. Additionally, a more comprehensive discussion on the value of HbA1c as a predictive measure for mortality is warranted, particularly in light of its moderate predictive power observed in the AUROC analysis.

As we move forward, it is crucial to evolve our clinical practices to ensure that prediabetic STEMI patients receive the comprehensive management they require, ultimately recognizing them as a high-risk group deserving of focused attention.

## Figures and Tables

**Figure 1 biomedicines-12-02223-f001:**
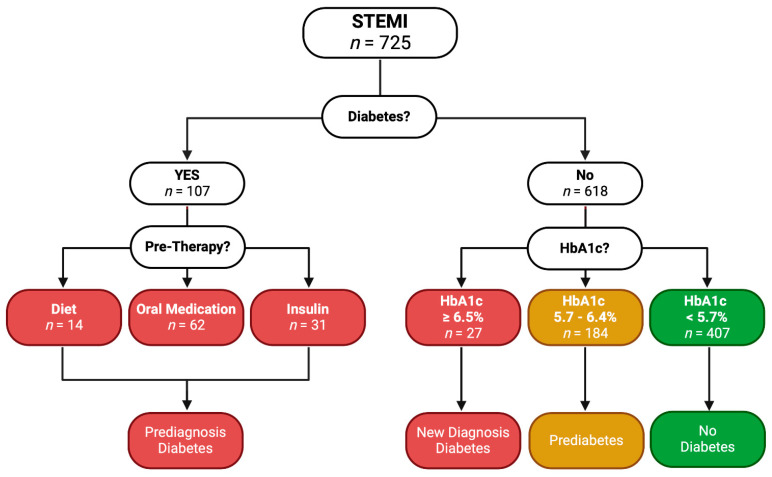
Flowchart of study design.

**Figure 2 biomedicines-12-02223-f002:**
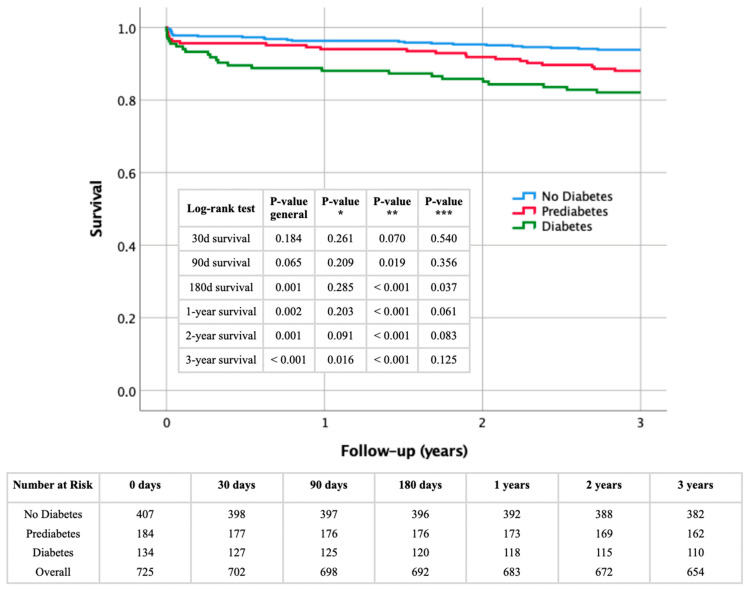
Kaplan–Meier survival analysis with corresponding numbers at risk and log-rank tests for detection of short- and long-term mortality regarding glycemic status in STEMI patients. *p** No diabetes vs. prediabetes; *p*** no diabetes vs. diabetes; *p**** prediabetes vs. diabetes.

**Figure 3 biomedicines-12-02223-f003:**
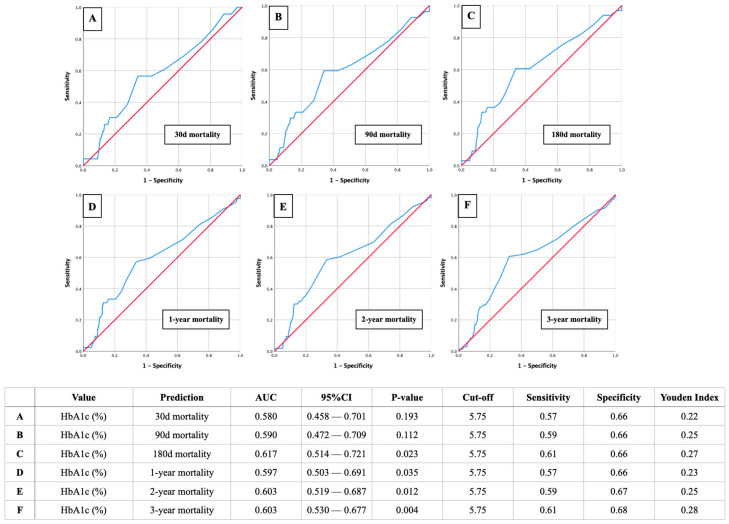
Area Under the Receiver Operating Characteristic (AUROC) analysis of HbA1c with corresponding cut-off values, sensitivity, specificity, and Youden index for short- and long-term mortality in patients with STEMI (Panel (**A**): 30d mortality; Panel (**B**): 90d mortality; Panel (**C**): 180d mortality; Panel (**D**): 1-year mortality; Panel (**E**): 2-year mortali-ty; Panel (**F**): 3-year mortality).

**Figure 4 biomedicines-12-02223-f004:**
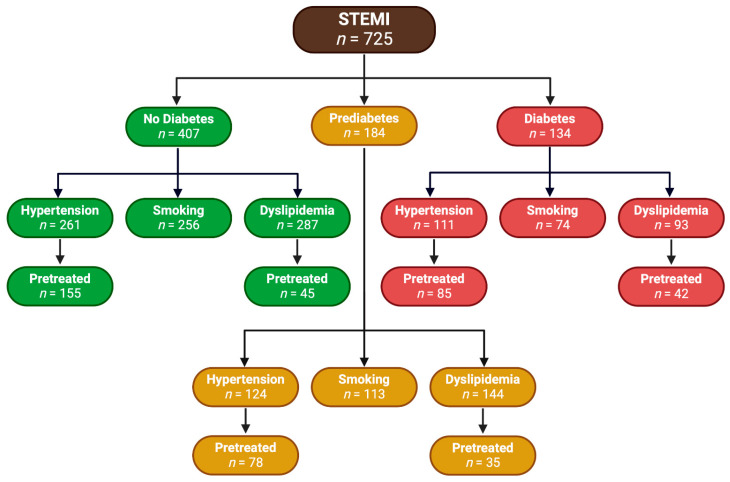
Glycemic status and further cardiovascular risk factor distribution of study cohort.

**Figure 5 biomedicines-12-02223-f005:**
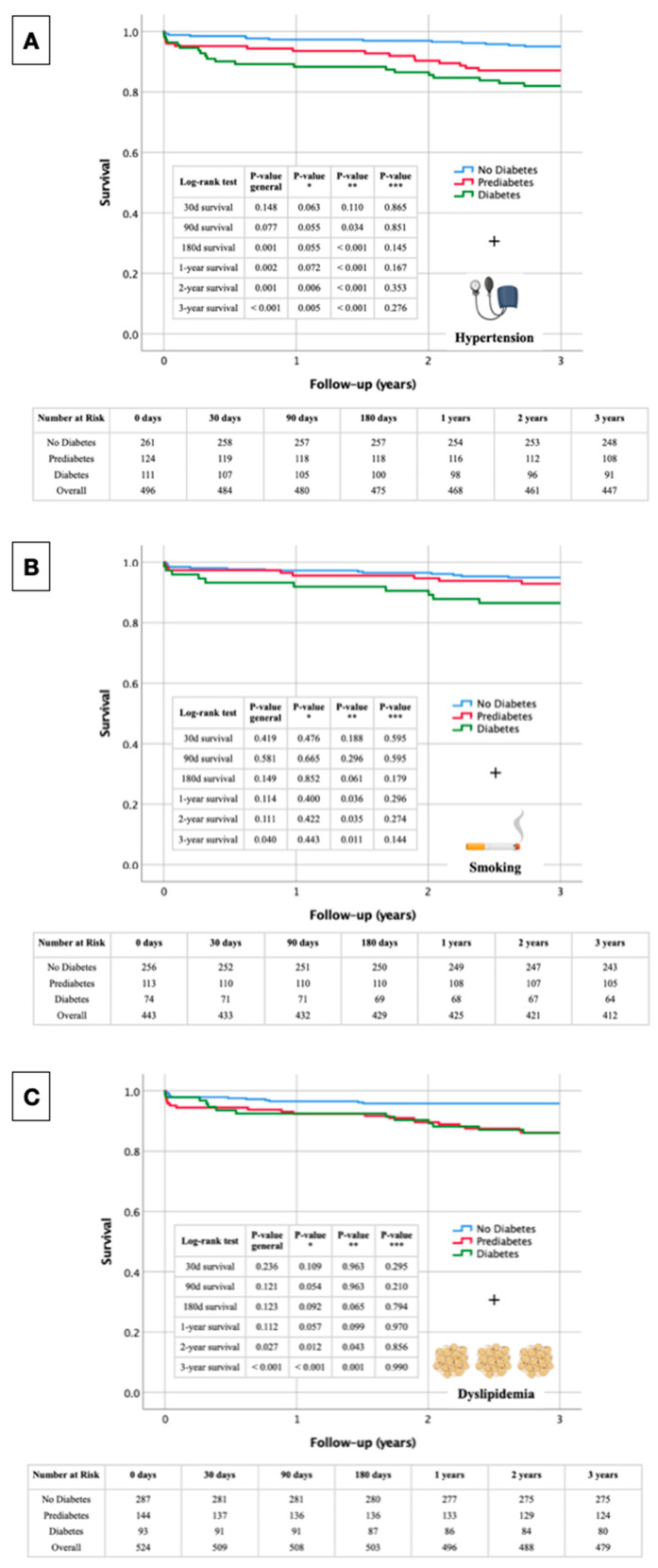
Kaplan–Meier survival analysis with corresponding numbers at risk and log-rank tests for detection of short- and long-term mortality regarding glycemic status and several cardiovascular risk factors in STEMI patients (Panel (**A**): Glycemic status and arterial hypertension; Panel (**B**): Glycemic status and smoking; Panel (**C**): Glycemic status and dyslipidemia). *p** No diabetes vs. prediabetes; *p*** no diabetes vs. diabetes; *p**** prediabetes vs. diabetes.

**Table 1 biomedicines-12-02223-t001:** Baseline characteristics of study cohort.

.	No Diabetes	Prediabetes	Diabetes	*p**	*p***	*p****
*n* (%)						
Number	407 (56.1)	184 (25.4)	134 (18.5)	-	-	-
Gender (Male)	301 (74.0)	130 (70.7)	100 (74.6)	0.403	0.878	0.434
Age						
<50 years	60 (14.7)	13 (7.1)	11 (8.2)	**0.009**	0.052	0.703
50–69 years	228 (56.0)	107 (58.2)	71 (53.0)	0.628	0.540	0.359
70–79 years	89 (21.9)	40 (21.7)	37 (27.6)	0.972	0.172	0.227
≥ 80 years	30 (7.4)	24 (13.0)	15 (11.2)	**0.027**	0.165	0.620
CVRF						
Arterial Hypertension	261 (64.1)	124 (67.4)	111 (82.8)	0.441	**<0.001**	**0.002**
Dyslipidemia	287 (70.5)	144 (78.3)	93 (69.4)	**0.050**	0.807	0.073
Smoking	256 (62.9)	113 (61.4)	74 (55.2)	0.730	0.114	0.268
BMI						
<20.0 kg/m^2^	9 (2.2)	6 (3.3)	5 (3.7)	0.448	0.320	0.802
20.0–24.9 kg/m^2^	137 (33.7)	47 (25.6)	19 (14.2)	**0.049**	**<0.001**	**0.013**
25.0–29.9 kg/m^2^	194 (47.6)	74 (40.2)	58 (43.3)	0.096	0.352	0.628
30.0–34.9 kg/m^2^	50 (12.3)	46 (25.0)	37 (27.6)	**<0.001**	**<0.001**	0.575
35.0–39.9 kg/m^2^	17 (4.2)	10 (5.4)	13 (9.7)	0.491	**0.013**	0.137
>40.0 kg/m^2^	0 (0.0)	1 (0.5)	2 (1.5)	0.136	**0.013**	0.380
STEMI Characteristics						
CHD–1 vessel	178 (43.7)	70 (38.0)	43 (32.1)	0.194	**0.017**	0.273
CHD–2 vessels	104 (25.6)	68 (37.0)	32 (23.9)	**0.005**	0.699	**0.013**
CHD–3 vessels	125 (30.7)	46 (25.0)	59 (44.0)	0.156	**0.005**	**<0.001**
CPR	55 (13.5)	16 (8.7)	18 (13.4)	0.095	0.981	0.177
Cardiogenic shock	62 (15.2)	18 (9.8)	28 (20.9)	0.073	0.127	**0.005**
Fibrinolysis	8 (2.0)	3 (1.6)	1 (0.7)	0.780	0.338	0.485
DES implantation	387 (95.1)	172 (93.4)	121 (90.3)	0.237	**0.028**	0.564
CABG indication	17 (4.2)	6 (3.3)	7 (5.2)	0.594	0.598	0.375
In-hospital death	8 (2.0)	7 (3.8)	10 (7.5)	0.188	**0.002**	0.152
Previous Disease						
MI	32 (7.9)	30 (16.3)	21 (15.7)	0.188	**0.008**	0.879
PCI or CABG	41 (10.1)	33 (17.9)	29 (21.6)	**0.002**	**0.001**	0.410
Heart Failure	12 (2.9)	11 (6.0)	14 (10.4)	**0.008**	**<0.001**	0.138
Renal Failure	28 (6.9)	18 (9.8)	31 (23.1)	0.078	**<0.001**	**0.001**
AF	14 (3.4)	11 (6.0)	11 (8.2)	0.223	**0.023**	0.439
Cancer	36 (8.8)	18 (9.8)	7 (5.2)	0.156	0.367	0.299
mean ± SD						
Age (years)	61.9 ± 12.2	64.7 ± 11.8	65.7 ± 10.9	**0.010**	**0.002**	0.464
BMI (kg/m^2^)	26.5 ± 3.9	27.9 ± 5.4	29.1 ± 4.8	**0.002**	**<0.001**	**0.044**
LVEF (%)	43.9 ± 8.9	42.5 ± 9.5	41.6 ± 11.3	0.086	**0.031**	0.427
median ± IQR						
Laboratory Values						
TC (mg/dL)	185.0 ± 63.5	186.0 ± 64.0	168.0 ± 63.5	0.949	**<0.001**	**0.001**
HDL (mg/dL)	50.0 ± 18.0	48.0 ± 21.0	41.5 ± 19.3	0.057	**<0.001**	**0.002**
Non-HDL (mg/dL)	134.0 ± 61.5	133.0 ± 65.0	126.0 ± 60.8	0.513	**0.024**	**0.014**
LDL (mg/dL)	111.0 ± 57.5	111.0 ± 60.0	93.0 ± 55.0	0.814	**<0.001**	**<0.001**
Triglycerides (mg/dL)	103.0 ± 69.0	112.0 ± 81.0	143.5 ± 108.3	0.078	**<0.001**	**<0.001**
Troponin T max (ng/L)	3504.0 ± 5347.0	3417.0 ± 4919.0	3220.0 ± 6415.0	0.974	0.847	0.848
Creatinkinase max (U/L)	1475.0 ± 2252.5	1298.0 ± 2154.0	1426.0 ± 2080.0	0.841	0.515	0.664
HbA1c (%)	5.4 ± 0.3	5.8 ± 0.3	7.3 ± 1.6	**<0.001**	**<0.001**	**<0.001**

*p** No diabetes vs. prediabetes; *p*** no diabetes vs. diabetes; *p*** prediabetes vs. diabetes; CVRF: cardiovascular risk factors; BMI: body mass index; STEMI: ST-segment elevation myocardial infarction; CHD: coronary heart disease; CPR: cardiopulmonary resuscitatio; DES: drug eluting stent, CABG: coronary artery bypass graft; MI: myocardial infarction; PCI: percutaneous coronary intervention; AF: atrial fibrillation; LVEF: left ventricular ejection fraction; TC: total cholesterol; HDL: high-density lipoprotein; LDL: low-density lipoprotein.

## Data Availability

The raw data supporting the conclusions of this article will be made available by the authors on request.

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
