# Peer review of "The Silent Threat: A Retrospective Cohort Study on the Impact of Prediabetes on ST-Elevation Myocardial Infarction Survival—A Call to Action!"

_biomedicines, 2024, doi:10.3390/biomedicines12102223_

Round 1

Reviewer 1 Report

Comments and Suggestions for Authors

This study addresses an important clinical issue, but significant revisions are necessary to enhance its clarity, validity, and robustness. The title, abstract, methodology, and conclusions should be refined to better reflect the study design and findings, and to ensure that the limitations—particularly regarding the diagnosis of prediabetes and the predictive value of HbA1c—are fully acknowledged and discussed.

Below is a more comprehensive review report addressing the various aspects of the manuscript:

1. Study Design in the Title

The title does not clearly reflect the study design, as I thought it was a review article in my first impression. Given that the study is retrospective in nature, it is important to explicitly mention this in the title to provide clarity to the readers. For example, the title could be revised to include “A Retrospective Cohort Study” to reflect the methodology used. Consider revising to something like: “A Retrospective Cohort Study on the Impact of Prediabetes on STEMI Outcomes.”

2. Abstract Revision

The abstract requires restructuring to achieve a better balance between its components. Currently, it lacks proportion, with an overemphasis on the introduction and background while key aspects such as methodology, results, and conclusions are underrepresented. A revision is necessary to ensure that the abstract provides a clear, concise summary of the study, with more emphasis on the results and conclusions.

For instance, mention the cohort size (725 patients), key findings from Kaplan-Meier survival analysis, and AUROC values for HbA1c as a mortality predictor.

3. Support for Conclusion

The conclusions drawn from the study are not fully supported by the results presented. While the study indicates that prediabetic patients are at increased risk, the assertion that “this paradigm shift in treating prediabetes as seriously as diabetes may significantly improve long-term survival outcomes” is not fully substantiated by the data.

More robust data analysis is required to support such a strong claim. Author could rephrase such as “e.g.: it may or might be associated and needed further validation” and suggesting the need for multi-marker strategies.

Also, the manuscript should provide additional discussion on the limitations of the predictive value of HbA1c, as mentioned in AUROC analysis, which showed moderate predictive power for mortality.

4. Diagnostic Criteria for Prediabetes

A major limitation of the study is the reliance on a single HbA1c measurement to diagnose prediabetes. Current guidelines suggest that more than one test should be conducted to confirm the diagnosis of prediabetes. Using only one HbA1c measurement may lead to misclassification, affecting the study’s validity.

Acknowledge this limitation more explicitly in the discussion section. Emphasize that guidelines generally recommend multiple tests to confirm the diagnosis of prediabetes, and discuss the potential impact of misclassification on study results.

5. Methods Section and Outcome Determination

The methods section is insufficiently detailed, particularly regarding the determination of outcomes. Key aspects such as the criteria used for classifying outcomes, including how cardiovascular events and survival rates were determined, need to be more thoroughly described.

6. Reporting of Crude Event Incidences

The manuscript should report the crude event incidence at each time point for the different groups (non-diabetic, prediabetic, diabetic) in order to provide a clearer understanding of how the groups compare over time. Include in the table or main-text that reports the crude incidences of cardiovascular events and mortality across these groups at different time point

 is crucial for readers to assess the progression of events in the study population and is currently missing.

7. Sensitivity and Specificity of HbA1c

As noted in the manuscript, the sensitivity and specificity of HbA1c in predicting long-term mortality are relatively low. This should be emphasized in the discussion, and the conclusions should be tempered accordingly. Overstating the predictive power of HbA1c could lead to misleading interpretations, especially given the study’s focus on cardiovascular risk in prediabetic patients.

Minor:

l  Replace “received ethical approval from” with “was approved by”.

l  “The data were then pseudo-anonymized and entered into an Excel database.” could be rephrased for clarity: “Subsequently, patient data were pseudo-anonymized and compiled into an Excel database for analysis.”

l  Inconsistent spelling of “prediabetes” and “pre-diabetes” should be standardized throughout the manuscript.

l  “The data were then pseudo-anonymized and entered into an Excel database.” could be rephrased for clarity: “Subsequently, patient data were pseudo-anonymized and compiled into an Excel database for analysis.”

Reviewer 2 Report

Comments and Suggestions for Authors

Dear authors, 

The manuscript entitled, " The Silent Threat: Prediabetes and Its Impact on Heart Attack Survival – A Call to Action!" is an interesting study with very effective information and idea of research. I however feel that overall presentation and explanation of the study can be improved if you consider to address the following comments in final version of your manuscript. 

Improve Glycemic Status Assessment: Using only a single HbA1c result to classify glycemia status might not accurately describe the variability of glycemic control over time. This would be done by doing more assessments or even maybe considering the use of CGM to get a better clearer and closer look at their glycemic status elicited over what the duration they had been exposed to this intervention or simple justification of use of 
single HbA1c result.

Consider Other Confounders: There might be the effect of other confounding variables such as atrial fibrillation, accelerated atherosclerosis, and HF phenotypes these are all factors that relate to cardiovascular disease and they may provide additional information about the risk profiles of this study population. Incorporating an examination of these factors in here could bolster your results.

Extended Follow-Up Time: While your three-year follow-up period stands as a good resource, it is still arguably too brief for a complete grasp of the long-term implications of prediabetes on survival. Increased follow-up duration would help provide better data on how the progression of cardiovascular risks advances in individuals with prediabetes, and lead to a better determination of the long-term prognosis among them.

Incorporate Lifestyle and Treatment Variations: Over the follow-up period, changes in treatment regiments, adherence of patient to medication as well as lifestyle modifications could have a considerable effect on outcomes. This could offer a more comprehensive view of how these aspects increase potential cardiovascular risks and effect patient management by discussing these differences, and its influence on your findings.

Emphasizing Controversial Findings: In the discussion section, focusing on any conflicting results (especially regarding prediabetes management) may be helpful. For instance, you could mention how the jury is still out on certain pharmacological agents that are typically used to treat prediabetes and recommend a more in-depth exploration of all the relevant research studies available.

Action Item for Healthcare Providers: Bolster the call to action by providing more precise strategies for healthcare providers on the inclusion of prediabetes management in routine cardiovascular care. This might involve more precise instructions for monitoring, lifestyle interventions or medication if necessary, then success turn work into useful real world clinical applications.

Broaden Study Design: Your results may not generalize if your study is limited by design to a single center and retrospective. Taking the study in a multi-center manner would help vary the patients, lead to more generalizable conclusions and once again apply beyond our own arena.

Define Inclusion and Exclusion Criteria Simplify your study population: The criteria for including/excluding will differ based on the nature of studies. It would be good to give a wider rationale for how you picked participants, and if biased occurred in the cohort and how it was controlled for so we have some idea of who this might apply too.

Subgroup Analyses: Include subgroup analyses to investigate whether or not outcomes differ based on age, sex and comorbidities. This might identify relevant heterogeneity of effect sizes for the effect of prediabetes on cardiovascular risks by specific patient populations and therefore, provide individualized information on disease management.

Best regards

Comments on the Quality of English Language

minor editing is needed 
